# A Virtual Assistant to Guide Early Postoperative Rehabilitation after Reverse Shoulder Arthroplasty: A Pilot Randomized Trial

**DOI:** 10.3390/bioengineering11020152

**Published:** 2024-02-02

**Authors:** José-María Blasco, Marta Navarro-Bosch, José-Enrique Aroca-Navarro, David Hernández-Guillén, Pau Puigcerver-Aranda, Sergio Roig-Casasús

**Affiliations:** 1Group in Physiotherapy of the Ageing Processes—Social and Healthcare Strategies, Department of Physiotherapy, University of Valencia, 46010 Valencia, Spain; jose.maria.blasco@uv.es (J.-M.B.); sergio.roig@uv.es (S.R.-C.); 2Orthopedic and Trauma Surgery Service, Hospital Universitari i Politècnic La Fe de València, 46026 Valencia, Spain; mnavarrob26@yahoo.es (M.N.-B.); joseenriquearoca@yahoo.com (J.-E.A.-N.); 3Department of Physiotherapy, University of Valencia, 46010 Valencia, Spain

**Keywords:** reverse shoulder arthroplasty, rehabilitation, chatbot, virtual assistant, smartphone

## Abstract

Introduction: Rehabilitation can improve outcomes after reverse shoulder arthroplasty (RSA). However, low adherence to rehabilitation and compliance rates are some of the main barriers. To address this public health issue, the goal of this research was to pilot test and evaluate the effectiveness of a chatbot to promote adherence to home rehabilitation in patients undergoing RSA. Methods: A randomized pilot trial including patients undergoing RSA and early postoperative rehabilitation was performed. The control group received standard home rehabilitation; the experimental group received the same intervention supervised with a chatbot, with automated interactions that included messages to inform, motivate, and remember the days and exercises for 12 weeks. Compliance with rehabilitation and clinical measures of shoulder function, pain, and quality of life were assessed. Results: 31 patients (17 experimental) with an average age of 70.4 (3.6) completed the intervention. Compliance was higher in the experimental group (77% vs. 65%; OR95% = 2.4 (0.5 to 11.4)). Statistically significant between-group differences with a CI of 95% were found in the QuickDASH questionnaire and self-reported quality of life. No differences were found in the rest of the measures. Conclusions: This pilot study suggests that the chatbot tool can be useful in promoting compliance with early postoperative home rehabilitation in patients undergoing RSA. Future randomized trials with adequate power are warranted to determine the clinical impact of the proposal.

## 1. Introduction

Reverse shoulder arthroplasty (RSA) is considered one of the most successful solutions in reconstructive shoulder surgery. The procedure was initially recommended for patients with rotator cuff arthropathy. Now, surgeons have expanded its application to massive cuff tears without arthritis, fracture care, rheumatoid arthritis, and failed prior surgery replacements with a high level of success [1,2]. The RSA procedure flips the normal shoulder ball and socket anatomy. This stabilization design results in a semi-constrained prosthesis that stabilizes the glenohumeral center of rotation analogous to a functioning rotator cuff.

Many efforts have focused on advancing the understanding of prosthesis biomechanics and testing surgical modifications. Despite remarkable benefits in terms of pain, function, and quality of life [3,4], there may be complications [5,6,7]. Optimal postoperative management is still debated; for instance, the impact of immobilization times and rehabilitation procedures on clinical outcomes [8]. In addition, there is no clear consensus on the rehabilitation guidelines. However, rehabilitation is considered essential to maximize the return to basic daily functional activities or even more complex activities, such as sports practice [8].

Several factors can contribute to the success of surgery, among which is adherence to rehabilitation [9,10]. In developed countries, postoperative rehabilitation care after major surgeries is usually implemented, which involves a period of inpatient—immediately after surgery—and outpatient supervised programs, along with education sessions for self-management. However, adherence to caregivers’ recommendations represents one of the main barriers to recovery [11]. The reasons for low adherence are aspects such as patient-centered support, patient and clinician beliefs, and overall therapeutic response. Other factors include previous low levels of physical activity, depression, anxiety, helplessness, poor social support/activity, a greater perceived number of barriers to exercise, and increased pain levels during exercise [12].

Therefore, adherence continues to be an important concern in rehabilitation, not only in cases of shoulder pain and surgery, but in overall musculoskeletal, neurological, and other types of disorders. Numerous contemporary studies are focused on testing the possible strategies to promote adherence. The literature on the topic is vast due to its clinical relevance [12]. However, the use of information and communication technology (ICT) has gained interest for its wide possibilities, although it is not exempt from challenges [13,14,15]. Monitoring can be conducted through telephone calls, pre-recorded videos, videoconferences, and virtual reality environments [16,17,18,19,20]; therefore, more digital solutions are used every day in daily activities, especially after the paradigm shift brought about by the COVID-19 pandemic, which spread around the world in 2020. Among technological solutions, smart mobile devices probably offer the greatest potential to promote adherence to remote care, mainly through mobile applications that are designed for phones and smart devices and enable tasks to be carried out, facilitating procedures or activities.

Current trends seem to advocate for the use of smart mobile devices to perform physical exercise and promote health, with a significant number of applications also in physiotherapy [21]. However, this study proposes the use of a chatbot instead. Chatbots are designed to simulate a human conversation. Algorithms are usually implemented from the design of flow diagrams or even decision trees, a prediction model that, based on predefined questions and answers, assists in programmed tasks. This is an interesting and innovative alternative that is currently being explored in various healthcare fields [22,23] and deserves exhaustive exploratory analysis in the field of rehabilitation because its use is simple and intuitive. Indeed, the use of mobile applications can represent an insurmountable barrier for individuals, such as older people, i.e., the need for access, updates, etc., even though mobile communication is widespread.

This study hypothesizes that the use of a chatbot that communicates via an instant messaging application on a smartphone could be an effective strategy to promote adherence and clinical outcomes in patients undergoing RSA when compared with usual care. The goal was to conduct a pilot trial to assess the effects of the proposed chatbot on adherence and clinical outcomes assessed at 12 weeks after RSA.

## 2. Materials and Methods

### 2.1. Design

This was a randomized pilot clinical trial conducted in patients undergoing RSA and was designed with two arms: an experimental group, whose participants underwent a postoperative rehabilitation program assisted with a chatbot, and a control group, whose participants underwent the same rehabilitation program without such an assistant. This study was conducted at the Hospital Universitari i Politècnic la Fe, (Valencia, Spain) between June 2022 and December 2023. The Universitat de València (Valencia, Spain) was responsible for the integrity and conduct of the research.

### 2.2. Participants, Randomization, and Masking

Patients on the waiting list for shoulder surgery were invited by the principal investigator to participate in this study if they complied with the following inclusion criteria: undergoing RSA within the next month; owning a personal smartphone with an instant messaging application installed, regardless of the manufacturer; familiarity with the use of such an application, and this criterion was verified by asking the patient “*Do you access the application more than three times per week?*”; and an ability to understand, write, and read in Spanish. Patients who presented with any condition other than those associated with shoulder pathology, whether cognitive, neurological, integrative, or musculoskeletal, had difficulty in either understanding the caregivers’ instructions or performing the exercises, or in whom the exercises might pose a risk to the patient’s health, were excluded. Once the inclusion criteria were verified, the patients were informed about the trial verbally and provided with an information brochure. If a patient agreed to participate, they had to sign a consent form.

A binomial randomization sequence was generated in origin with computer software (Matlab^®^ 9.13.0 (R2022b)). Patients were enrolled sequentially and assigned to the experimental or the control group by the principal investigator, according to the sequence output. An orthopedic surgeon with more than 15 years of experience, blinded to patient assignment, was in charge of conducting the surgeries. A member of the research team, a physiotherapist with more than 15 years of experience, was informed about the patients’ assignment and was in charge of educating patients in the rehabilitation program and supervising the interventions, as described later. The researcher in charge of physiotherapy interventions was not blinded. Another researcher, blinded to group allocation, was in charge of assessing all participants. The extracted data were coded by the principal investigator and analyzed by the biometrician of the current study.

### 2.3. Interventions

#### 2.3.1. Surgical Procedure

General anesthesia was used, associated with locoregional anesthesia, by inter scalene brachial plexus block. The patient was placed in the beach chair position. Dislocation of the humeral head was achieved through a combination of adduction, external rotation, and retropulsion maneuvers while applying pressure on the elbow from below to above. The posterior rotator cuff was then checked, as were the remains of the infraspinatus and the insertion of the teres minor. Humeral head resection was performed using a central-medullary cutting guide, slightly below the top of the greater tuberosity. At this stage, a protector was placed on the bone cut to prevent it from being deformed by the pressure of the retractors while the glenoid was being prepared. Extensive circumferential glenoid capsulotomy and labrum resection were then performed. Glenoid exposure should allow free choice of the entry point of the drill guide and the direction of the central peg for the implantation of the base plate. It was important to avoid positioning the center hole too high due to the risk of inferior bone overlap (which can generate scapular notching). The glenoid was then reamed, but not excessively so as to preserve solid subchondral bone. The central peg was then prepared. Impaction of the metagene was performed, and the final attachment of the baseplate was obtained by screw fixation; after that, the glenosphere was assembled. Once the glenoid implant was in place, the surgeon subluxated the humerus superiorly and anteriorly. Two types of reamers were needed: one for the medullary canal and the other for the metaphyseal zone. The chosen degree of retroversion was carefully preserved during the preparation of the humerus. The humeral component was then positioned, and the prosthesis was reduced. The height of the metaphyseal element and the polyethylene insert were chosen by careful testing of muscular tension. The incision was closed on a redon drain. Special care had to be taken when closing the deltoid during a superolateral approach (with trans-osseous sutures in the acromion) or when suturing the subscapularis during a deltopectoral approach. Immobilization was obtained using a sling.

#### 2.3.2. Rehabilitation Program

After surgery, patients spent an average of 2 days hospitalized, during which they received a first visit from the physiotherapist. Approximately one week after the surgical procedure, the participants were scheduled to have a face-to-face session with the physiotherapist in charge of the interventions. It was decided to design a rehabilitation program in which participants began performing moderate exercises early after surgery, as discussed below. The first education session aimed to explain the exercises that should be performed from the day after the appointment until 12 weeks after surgery, with a total of 48 to 50 scheduled sessions. The program could be carried out at a time that was convenient for the patient, with an indication of completion every weekday. The rehabilitation program was the same for all participants. The difference between the groups was as follows: in the education session, patients in the control group received a brochure with a description of the exercises, along with a calendar designed for this project to record the sessions carried out. Approximately one month after this session, these patients were scheduled for a second education session, oriented around supervising how the exercises were performed and solving possible doubts. In the session, patients were reminded to fill out the calendars and register the completed days.

On the other hand, the participants in the experimental group were instructed on how to use the chatbot tool. They were advised to receive automatic instant messages at the agreed time to perform the exercises. Such messages included information about the specifics of the intervention, such as its length or the number of sessions per week. In addition, every session day, the participants were inquired about whether they were going to perform the exercises; if so, the session was recorded, and the multimedia material designed for this study with the exercises and repetitions to be performed was recorded. Finally, the program was designed with follow-ups at the end of each week on aspects such as pain or perceived difficulty and reports on progress and compliance.

### 2.4. Measures

#### 2.4.1. Adherence

Adherence to rehabilitation was assessed by registering the number of completed physiotherapy sessions, *n*, and the rate between *n* and the total programmed sessions. Adherence was considered to be achieved for participants who complied with at least 80% of the program [24,25]. Participants in the control group were provided with a calendar and were instructed to register every training day and time. One month after surgery, participants were contacted and reminded to fill in the calendars. Participants in the experimental group registered this outcome electronically by answering predefined questions in the chatbot.

#### 2.4.2. Clinical Assessment

Patients were measured at baseline, i.e., before undergoing surgery, and three months after surgery. The following measures were assessed:-Shoulder disability was assessed with the QuickDASH questionnaire. This is a self-reported survey of 11 items designed to assess symptoms and the ability to perform certain activities. The participants were instructed to answer every question based on their condition in the last week by circling the appropriate number. If participants had not had the opportunity to perform an activity in the past week, they were instructed to make their best estimate of which response was the more accurate, based on their ability regardless of how the task was performed. At least 10 items had to be answered to calculate a score. Each answer could be scored from one to five, and the average value was calculated. Then, to express the score in percentages, 1 was subtracted from the result, which was then multiplied by 25. The higher the score, the greater the disability. This questionnaire has been validated in the Spanish language (ICC = 0.8) [26], with a minimal clinically important difference MCID95% = 20 points.-Shoulder pain was assessed with a numeric pain rating scale, ranging from 0, the worst possible pain, to 10, no pain (MCID95% = 1.1 to 2.17) [27,28,29]. Pain was assessed with the question: What was your overall pain in the shoulder in the last week?-Shoulder functionality, as assessed with the Constant score, is a multi-item and summative scale that provides a global score based on weighted measures of physical impairments in the range of motion and strength, along with patient-reported pain and activity limitation of the affected shoulder. The scores ranged from 0 to 100 points, representing the worst and best shoulder function, respectively. The test is divided into four subscales: pain (15 points), activities of daily living (20 points), strength (25 points), and range of motion, i.e., forward elevation, external rotation, abduction, and internal rotation of the shoulder (40 points). The MCID95% = 10.4 [30].-Quality of life was measured with the EQ-5D-5L questionnaire, which essentially consists of 2 pages: the EQ-5D descriptive system and the EQ visual analog scale (EQ VAS). The descriptive system comprises five dimensions: mobility, self-care, usual activities, pain/discomfort, and anxiety/depression. The EQ VAS records the patients’ self-rated health on a vertical visual analog scale, where the endpoints are labeled “The best health you can imagine” and “The worst health you can imagine”.-The grade of difficulty perceived when exercising was assessed with a 0 to 5 Likert scale of difficulty.-Pain when exercising was assessed with a 0 to 5 Likert scale of pain.-Patient satisfaction with (1) the use of the tool (only the experimental group) and (2) the surgery was assessed with a 0 to 10 scale.

### 2.5. Data Analysis

A descriptive synthesis of the characteristics of the patients was conducted, using frequencies, means, standard deviations, and contingency tables. Inferential analysis was conducted with IBM SPSS statistics software 24.0 licensed by the Universitat de València to assess the effectiveness of interventions and compare clinical outcomes. Baseline status was compared with unpaired t-tests. To test the study hypothesis, an analysis of the variance of repeated measures was conducted to find possible time (baseline to postintervention), group (experimental vs. control), and time × group interactions. Tukey and Dunnet tests were planned for post hoc comparisons. The odds of becoming adherent when receiving the experimental intervention against receiving standard (control) care were estimated considering positive events, i.e., adherence achieved when patients complied with over 80% of the rehabilitation program, and vice versa. All confidence intervals were set at 95%. This was a pilot study that was designed in consideration of the feasibility of the proposal; thus, formal a priori sample size calculation was not undertaken. The aim was to recruit 30 patients; this sample size being a pragmatic decision based largely on general recommendations advocated for pilot studies, as well as on aspects such as the study aim and the availability of resources [31].

### 2.6. Ethics Statement

This study was designed to adhere to the clinical and ethical guidelines set in the Declaration of Helsinki. To ensure compliance with ethical standards, the Comité de Ética de la Investigación con medicamentos, CEIm, at the Hospital La Fe (CPMP/ICH/135/95) ruled that the project complied with the ethical regulations on biomedical research with human subjects and was viable in terms of the scientific approach, objectives, material, and methods, as well as the patient information sheet and in the informed consent and current protection data laws (number of registries: 2021-931-1). All participants were informed verbally and in writing about the specifics of the project and their participation and signed a consent form to participate. Participants were allowed to withdraw from the study at any time, without prejudice to their medical care.

## 3. Results

Overall, 31 patients were recruited according to the criteria and were allocated to the experimental (*n* = 17) or the control group (*n* = 14) and completed the intervention. The patients were mainly women (21 women, 10 men) with an average age of 70.4 (3.6) years old. Approximately half the participants reported performing some physical exercise regularly, such as walking (13/30, 43%), cycling (4/30, 13%), or Pilates (1/30, 3%). A flowchart of participants is shown in Figure 1, and the baseline characteristics are presented in Table 1.

### 3.1. Adherence

The average adherence in the experimental group was 77%, while in the control group, it was 65%, and the odds were OR95% = 2.4 (0.5 to 11.4; *p* = 0.128). However, four patients in the control group did not bring the calendar to the postoperative evaluation, so if these patients were to be considered in the overall calculation, adherence fell considerably below the referred value, and the odds of being adherent with the tool against standard care increased to OR95% = 8.1 (1.6 to 40.7; *p* = 0.005). Specifically, 12/17 (70%) participants in the experimental group were classified as achieving adherence goals to rehabilitation, i.e., the completion of more than 80% of programmed sessions, whereas 8/14 (58%) participants achieved this goal in the control group. Almost 20% of participants reported that exercising was very difficult; the rest distributed the performance of exercises between difficult and normal in an equal manner. Most patients (90%) reported some level of pain while exercising, although in most cases (58%) the pain was described as moderate, as is discussed subsequently.

### 3.2. Clinical Effects of Intervention

The results shown in Table 2 suggest that participants evolved with significant benefits in terms of shoulder function and shoulder disability, pain, and quality of life (all of them with *p* < 0.001). However, there were no significant increases in shoulder strength or mobility. Only one of the five shoulder range of motion measures increased significantly: shoulder flexion. As for group interactions, the 45-point reduction in shoulder disability in participants of the experimental group, according to the QuickDASH score, overcame the minimal detectable change and was significantly higher than the 39-point reduction registered in the control group. No other noticeable group difference was found. However, some residual results suggested that the increase in the perceived quality of life was higher in the experimental group, although only in terms of the visual analog scale evaluation. In addition, the increase in the shoulder adduction range of motion suggested a similar outcome. Overall, there were no time × group interactions, as shown in Table 1, suggesting similar effects of interventions.

## 4. Discussion

This study indicated that those participants undergoing RSA who received one session of education and a domiciliary physiotherapy treatment supervised with a chatbot via an instant messaging smartphone application achieved greater compliance rates to rehabilitation than those participants who underwent standard care based on the same physiotherapy program but did not use such an application. The main finding is that the chatbot proposal can be an effective strategy to increase adherence to early postoperative rehabilitation in patients undergoing RSA.

Achieving a positive impact on adherence mainly intends to promote surgical and clinical outcomes. The results suggested that all participants exhibited significantly reduced shoulder disability and increased shoulder functionality in the early postoperative period, in terms of the QuickDASH and Constant scores. There were clinically important improvements that exceeded the respective 20 and 10.4 points established for such questionnaires [26,27,29,30]. In addition, a higher rate of adherent participants in the experimental group resulted in greater clinical benefits for the participants; these improvements were significant in statistical terms, which supported the effectiveness of the proposal, which was also in clinical terms.

The effects of the rehabilitation program on shoulder function in terms of strength and range of motion were also positive. Participants in both groups exhibited a significantly improved range of motion in shoulder flexion; however, the experimental group increased their shoulder adduction range to a higher extent. The results appeared to indicate that such benefits may or may not be due to the implementation of one intervention or another. The reason for such a statement is that five shoulder movements were evaluated, but for the 5 × 2 possibilities of between-group or pre-to-post intervention differences, only two interactions with significant differences were found. The fact that the specific benefits in flexion and adduction motion could have a direct cause–effect relationship with the use of the experimental digital tool requires further investigation and is open to speculation. In addition, the results suggested increased postintervention shoulder flexion strength, but the increase was not important in clinical terms, and no between-group differences were revealed.

Pain is a critical outcome and one of the main reasons why patients decide to undergo surgery when conservative treatments, such as analgesic drugs, do not resolve clinical symptoms. RSA is an effective solution to reduce pain [5], as supported in our study, with an approximate average reduction of six points, and this was regardless of group allocation. Pain reduction was clinically important since it exceeded the 2.17 upper limit suggested by some previous studies in a numerical pain rating scale oriented to the shoulder joint [27,28,29]. This positive result also implied that there were no differences between groups; therefore, our findings did not allow us to confirm whether one intervention was superior to the other in reducing pain.

Overall, RSA is considered a cost-effective and successful solution that increases the quality of life of patients [32,33]. Our results supported such a view since the patients’ quality of life improved in a short period. Studies have also indicated that in the medium and long term, this result could be even more favored as long as there are no complications and the survival of the prosthesis is adequate [33]. Although our results suggested that the use of the chatbot could be a differential factor in increasing the self-perceived quality of life to a greater extent than the standard treatment, the results of the questionnaire suggested that the effects were similar, regardless of the group. A longer follow-up time could enable this outcome to be exhaustively analyzed; the lack of this was a limitation of this study.

Despite the increasing use of RSA, the optimal postoperative management regimen ensuring the best patient outcomes is still debated. The literature regarding the various possibilities of rehabilitation is heterogeneous in methods and quality, and scant high-quality evidence guides current rehabilitation protocols [8]. For instance, there is no clear consensus regarding the type of immobilization, the timing of rehabilitation, or the convenience of supervised versus home programs. In contrast to some immobilization recommendations, in this research, we applied a domiciliary protocol aimed at performing early and progressive mobilizations, which, according to the early postoperative results, seemed to have been effective.

It is necessary to qualitatively emphasize some aspects associated with this approach. For example, most patients reported some level of pain when performing the exercises in the first weeks. Although the pain was mostly described as moderate and was reduced in progressive weeks, it could have directly influenced adherence ratios. Indeed, the literature indicates that pain is one of the main barriers to exercise [12]. Consistently, the individual analysis of the patients’ views allowed us to verify that, in most cases, a lower adherence ratio was associated with higher reported levels of pain when performing the exercises. The level of satisfaction with the rehabilitation program was high, although four patients (15% of the sample) reported feeling dissatisfied or very dissatisfied. A detailed analysis of the reasons verified that these patients were those who felt the highest levels of pain when exercising. Therefore, the level of early postoperative pain is one of the main aspects that caregivers should consider in postoperative management.

As for the strengths of this study, some points are worth considering. Firstly, the results were positive in terms of adherence. Secondly, the positive impact of adherence translated into some important clinical outcomes, such as benefits for shoulder disability and perceived quality of life. In contrast, the tool did not seem to promote other aspects, such as shoulder strength or range of motion, over standard care. On the other hand, chatbots are a contemporary and groundbreaking technology. The possible fields of application encompass different industries. They are increasingly present in our daily lives, and their involvement in daily life is expected to increase. In some healthcare fields, they have already been explored [34], but their use to guide the treatment of musculoskeletal disorders is still scarce. This study represents the first time that this proposal has been evaluated in patients with RSA. The clinical applicability of the proposal is worth highlighting. Chatbots allow rehabilitation programs to be prescribed, whether tailored or standard, in a short time, and work automatically. This facilitates patient monitoring with few health resources, especially in terms of personnel, which can considerably reduce the current economic burden.

This study offers a critical discussion of findings, a description of potential clinical impacts and applications, and contextualization with the contemporary literature. The results have been positive in terms of adherence and shoulder functionality, but some limitations should be acknowledged. The results were evaluated in the early postoperative period, so further follow-up evaluations, preferably after one year of surgery, would have enabled a better vision of the effects in the medium term when plateau values are expected to have recovered. The tool allowed compliance to be recorded electronically. However, an important aspect is that some of the patients in the control group did not bring the calendars that were provided to them to the follow-up evaluation session. The management of the outcomes of these patients is, therefore, debatable. Being methodologically strict, it must be interpreted that not submitting the schedule possibly indicates noncompliance with the program. One of the main limitations is that this was a pilot study with a limited sample size; therefore, the results should always be interpreted with caution. The results of this pilot study can guide the design of future research, and further clinical trials with adequate power are warranted to support the findings. These could be oriented to solve some of the mentioned limitations, with longer follow-up periods and a larger sample size.

## 5. Conclusions

Overall, this study suggests that a virtual assistant that interacts as a chatbot via an instant messaging smartphone application can be an effective strategy to promote adherence and increase rates of compliance to early postoperative home rehabilitation in patients undergoing RSA. Longer follow-up periods of 6 months to 1 year and a larger sample size could help to ultimately confirm such a statement and give clinical importance to the findings. The positive results of this pilot trial can help the design of future randomized clinical trials adequately powered to determine the short- and long-term clinical impact.

## Figures and Tables

**Figure 1 bioengineering-11-00152-f001:**
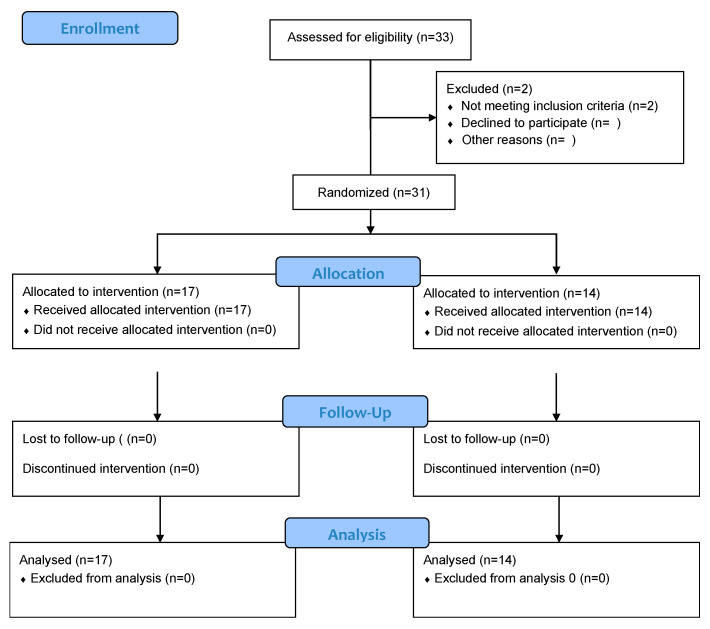
Flowchart of participants.

**Table 1 bioengineering-11-00152-t001:** Baseline characteristics of participants.

	Experimental (*n* = 17)	Control (*n* = 14)	Total (*n* = 31)	*p*-Value
Characteristics				
Age (years)	70.0 (4.2)	70.75 (3.8)	70 (73.4)	0.705
Weight (kg)	73.4 (16.1)	66.4 (13.5)	68.9 (14.8)	0.305
Affected shoulder (right, %)	71%	46%	58%	
Clinical assessment				
Shoulder function				
QuickDASH (score)	76.8 (17.3)	68.2 (15.1)	72.7 (16.4)	0.172
Constant (score)				0.188
Pain (NPRS 0 to 10)	8.2 (0.4)	7.9 (2.0)	8.1 (1.4)	0.661
Strength (flexion, kg)	2.6 (2.8)	3.2 (2.7)	2.9 (2.7)	0.647
Shoulder range of motion				
Flexion (degrees)	97 (15)	84 (25)	90 (21)	0.476
Extension (degrees)	22 (9)	22 (18)	22 (14)	0.778
Abduction (degrees)	77 (8)	65 (14)	71 (13)	0.504
Adduction (degrees)	13 (10)	18 (8)	18 (5)	0.783
External rotation (degrees)	29 (10)	31 (16)	37 (20)	0.176
Quality of life				
EQ-5D-VAS (score)	0.3 (0.1)	0.3 (0.3)	0.3 (0.2)	0.995
EQ-5D_auto (score)	34 (24)	50 (28)	41 (26)	0.117

Data are given as the mean (SD). *p*-values of *t*-tests were used for between-group basal assessments.

**Table 2 bioengineering-11-00152-t002:** Clinical effects of interventions.

	Experimental (*n* = 17)		Control (*n* = 14)		Total (*n* = 31)		Inferential Analysis		
	Post Intervention	Baseline Change	Post Intervention	Baseline Change	Post Intervention	Baseline Change	*p*-Value (Time)	*p*-Value (Group)	*p*-Value (Time × Group)
Shoulder function									
QuickDASH (score)	23.8 (9)	−45 (−66%)	38.6 (12.8)	−39 (51%)	31.6 (13.3)	−42 (−58%)	<0.001	0.048	0.351
Constant (score)	63.5 (12.6)	35.7 (128%)	58.4 (15.3)	23.3 (66%)	60.8 (13.9)	29.2 (92%)	<0.001	0.812	0.065
Pain (NPRS 0 to 10)	1.9 (1.4)	−5.9 (−76%)	2.2 (2.0)	−6.0 (−73%)	2.1 (1.7)	−6.0 (−74%)	<0.001	0.536	1.000
Strength (flexion, kg)	5.1 (2.3)	1.9 (59%)	4.2 (2.7)	1.6 (62%)	4.7 (2.5)	1.8 (62%)	0.07	0.451	0.845
Shoulder range of motion									
Flexion (degrees)	123 (19)	39 (46%)	107 (5)	10 (10%)	115 (21)	25 (28%)	0.002	0.867	0.039
Extension (degrees)	34 (17)	12 (55%)	29 (10)	7 (32%)	32 (14)	10 (45%)	0.14	0.504	0.644
Abduction (degrees)	78 (11)	13 (20%)	72 (3)	−5 (−6%)	76 (2)	5 (7%)	0.468	0.693	0.118
Adduction (degrees)	29 (2)	11 (61%)	16 (3)	3 (23%)	20 (11)	2 (11%)	0.272	0.01	0.056
External rotation (degrees)	34 (20)	3 (10%)	45 (15)	16 (55%)	32 (14)	5 (14%)	0.279	0.413	0.109
Quality of life									
EQ-5D-VAS (score)	0.7 (0.2)	0.3 (0.3)	0.7 (0.2)	0.3 (0.3)	0.7 (0.2)	0.3 (0.3)	<0.001	0.796	0.447
EQ-5D_auto (score)	81 (12)	31 (62%)	62 (20)	28 (82%)	71 (20)	30 (73%)	<0.001	0.041	0.773

Data are given as the mean (SD). Baseline change was calculated as the difference between postintervention and baseline assessments. Percentage change was calculated as PC = (Postintervention-Baseline)/Baseline. *p*-values of analysis of variance of repeated measures assessing time, group, and time × group interactions are shown.

## Data Availability

Data available on request due to privacy restrictions. The data presented in this study are available on request from the corresponding author.

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
