# Peer review of "A Virtual Assistant to Guide Early Postoperative Rehabilitation after Reverse Shoulder Arthroplasty: A Pilot Randomized Trial"

_bioengineering, 2024, doi:10.3390/bioengineering11020152_

Round 1

Reviewer 1 Report

Comments and Suggestions for Authors
  1. Please interchange the sentences in line #12 for better clarity.
  2. The term 'adherence' ('compliance') from line #18 should be introduced in the first instance of 'adherence' in line #12."
  3.  In line #30, there is an extra space before the word 'initially' that needs to be removed.
  4.  Line #30 is too long and the context is difficult to understand. Could you please rewrite it for clarity?
  5.  Line #33 begins with 'An RSA?'
  6.  Before mentioning 'an RSA' in line #33, there should be an introductory sentence explaining the usage of RSA.
  7. I suggest that the content of line #38 would be more appropriately started from line #37.
  8. The sentence in line #39 is overly long. Consider starting a new sentence from 'optimal postoperative' for better readability.
  9. In line #43, it is unclear which guidelines are being referred to and what activities to return to. Could you please specify?
  10. Please rewrite the sentence in line #45, as it is unclear.
  11. The sentence in line #50 is excessively long and needs revision for clarity.
  12. Mobile applications, or 'apps', are not the same?
  13. In line 67: 'From current trends, it seems appropriate to start a new paragraph
  14. In line 78: Given the factors mentioned above, the language should be revised for a more formal tone
  15. The sentence in line #104 is overly lengthy. It would benefit from being broken down into smaller, clearer sentences.
  16. The sentence in line #124 could be altered to enhance the understanding and clarity of the content.
  17. The phrase 'of this research' in line #128 could be changed to 'of the current study' or a similar term for a more formal tone.
  18. On the other hand, the phrase in line #173 should start a new paragraph.
  19. The sentence starting with 'Secondly' in line #177 should be revised for clarity.
  20. In line 192, the clinical assessment could be more clearly explained with the help of a diagram or table.
  21. In line 228, the word 'first' is unnecessary and can be removed.
  22. Line 245: In the results section, there is no information provided about the male sample.
  23. The last sentence in line #274 is too long and could be broken down into shorter, more understandable sentences.
  24. The last sentence in line #324 does not effectively conveys its intended context.
  25. In line #325, should it be 'studies' or 'study'?
  26. In line #340, the sentence initially starting with 'now' could be the beginning of a new paragraph.
  27. In line #363, the phrase 'this provide this research' does not appear appropriate in formal terms.
  28. Line 367: 'Current burden'—what exactly does this refer to?
  29. The last sentence in line #377 is unclear.
  30. The last sentence in line #381 does not appear appropriate in formal terms.
  31. The last sentence in line #385 could be revised for clarity.
  32. Conclusions? Please rewrite the entire conclusion section as it is not effectively presented

Comments on the Quality of English Language

already elaborated in the main comment section

Author Response

Comments to the reviewer:

Thank you for the time spent in making a thorough review of our manuscript. We have studied your comments and suggestions. They are reasonable and we believe that they can increase the quality of the manuscript if we address them correctly. In the following lines, you may find a point-by-point response to address the issues and suggestions

Reviewer Comments and Suggestions for Authors:

  1. Please interchange the sentences in line #12 for better clarity.
    • We have elaborated the abstract to better summarize the design, findings and conclusions
  2. The term 'adherence' ('compliance') from line #18 should be introduced in the first instance of 'adherence' in line #12."
    • Ok, done, both term are present in the new submission
  3. In line #30, there is an extra space before the word 'initially' that needs to be removed.
    • Amended
  4. Line #30 is too long and the context is difficult to understand. Could you please rewrite it for clarity?
    • The sentence has been split into two sentences, thank you
  5. Line #33 begins with 'An RSA?'
    • Changed: “The RSA procedure flips the normal shoulder ball and socket anatomy.”
  6. Before mentioning 'an RSA' in line #33, there should be an introductory sentence explaining the usage of RSA.
    • The procedure was initially recommended for patients with rotator cuff arthropathy. Now, surgeons have expanded its application to massive cuff tears without arthritis, fracture care, rheumatoid arthritis, and failed prior surgery replacements with a high level of success
  7. I suggest that the content of line #38 would be more appropriately started from line #37.
    • Ok, amended
  8. The sentence in line #39 is overly long. Consider starting a new sentence from 'optimal postoperative' for better readability.

The sentences have been shortened. The paragraph reads now: “Optimal postoperative management is still debated; for instance, the impact of immobilization times and rehabilitation procedures on clinical outcomes [8].

  1. In line #43, it is unclear which guidelines are being referred to and what activities to return to. Could you please specify?
    • Indeed: “In addition, there is no clear consensus on rehabilitation guidelines. However, rehabilitation is considered essential to maximize return to basic daily functional activities or even more complex activities such as sports practice [8].”
  2. Please rewrite the sentence in line #45, as it is unclear.
    • OK: “Several factors can contribute to the success of the surgery, among which is adherence to rehabilitation [9,10].”
  3. The sentence in line #50 is excessively long and needs revision for clarity.
    • Split into two sentences: “The reasons for low adherence are aspects such as patient-centered support, patient and clinician beliefs, and overall therapeutic response. Other factors include previous low levels of physical activity, depression, anxiety, helplessness, poor social support/activity, a greater perceived number of barriers to exercise, and increased pain levels during exercise [12].”
  4. Mobile applications, or 'apps', are not the same?
    • Indeed, apps removed
  5. In line 67: 'From current trends, it seems appropriate to start a new paragraph
    • Ok, changed
  6. In line 78: Given the factors mentioned above, the language should be revised for a more formal tone
    • The paragraph starts now with: “the goal of […]”
  7. The sentence in line #104 is overly lengthy. It would benefit from being broken down into smaller, clearer sentences.
    • Amended
  8. The sentence in line #124 could be altered to enhance the understanding and clarity of the content.
    • Done: “The researcher in charge of physiotherapy interventions was not blinded.”
  9. The phrase 'of this research' in line #128 could be changed to 'of the current study' or a similar term for a more formal tone.
    • Ok, amended
  10. On the other hand, the phrase in line #173 should start a new paragraph.
    • Amended
  11. The sentence starting with 'Secondly' in line #177 should be revised for clarity.
    • The whole paragraph has been revised and rewritten for clarity, thank you
  12. In line 192, the clinical assessment could be more clearly explained with the help of a diagram or table.
    • The whole section has been enumerated with dashes for clarity, hopefully, it is more readable now, thank you.
  13. In line 228, the word 'first' is unnecessary and can be removed.
    • Amended
  14. Line 245: In the results section, there is no information provided about the male sample.
    • “The patients were mainly women (21 women / 10 men) and aged an average of 70.4 (3.6) years old”
  15. The last sentence in line #274 is too long and could be broken down into shorter, more understandable sentences.
    • Ok, amended
  16. The last sentence in line #324 does not effectively conveys its intended context.
    • Indeed
  17. In line #325, should it be 'studies' or 'study'?
    • Changed by ‘research’
  18. In line #340, the sentence initially starting with 'now' could be the beginning of a new paragraph.
    • Amended
  19. In line #363, the phrase 'this provide this research' does not appear appropriate in formal terms.
    • Indeed, he sentence has been removed for clarity
  20. Line 367: 'Current burden'—what exactly does this refer to?
    • Economic burden è Changed
  21. The last sentence in line #377 is unclear.
    • It was confusing, we have decided to remove it since the information was not important
  22. The last sentence in line #381 does not appear appropriate in formal terms.
    • Same commentary as above, removed for clarity
  23. The last sentence in line #385 could be revised for clarity.
    • Rewritten for clarity: “However, the findings can guide the design of new research, and warrant future clinical trials with adequate power to support the findings”
  24. Conclusions? Please rewrite the entire conclusion section as it is not effectively presented
    • OK, the conclusions read now: “Overall, this study suggested that a virtual assistant that interacts as a chatbot via an instant messaging application in a smartphone can be an effective strategy to promote adherence and increase compliance rates to early postoperative home rehabilitation in patients undergoing RSA. Longer follow-up periods of 6 months to 1 year and a higher sample size could help to ultimately confirm such a statement and give clinical importance to the findings. The positive results of this pilot trial can help the design of future randomized clinical trials adequately powered to determine the short- and long-term clinical impact.”

Reviewer 2 Report

Comments and Suggestions for Authors

Rehabilitation can improve outcomes after reverse shoulder arthroplasty. However low adherence rates are one of the main barriers.

The goal of the AUTHORS was to evaluate the effectiveness of a Chatbot in supervising home rehabilitation and promoting adherence.

THEY applied a Randomized pilot trial at Hospital la Fe in Valencia, with patients undergoing reverse shoulder arthroplasty and early postoperative rehabilitation.

The control group received standard home rehabilitation; the experimental group received the same intervention supervised with a Chatbot, with automated interactions that included messages to inform, motivate and remember the days and training exercises for 12 weeks. The effects on adherence (compliance) and clinical measures of shoulder function, pain, and quality of life were assessed.

THEIR RESULTS highlight the following: 31 patients (17 experimental) aged 70.4 (3.6). Adherence was higher in the experimental group (77% vs. 65% of adherent participants; OR95%=2.4 (0.5 to 11.4)). Between-group differences were found in the Quick DASH and self-reported quality of life. No differences were found in the rest of the measures. Conclusion:

AUTHORS concluded that:- THEIR This pilot study suggests that the Chatbot tool is useful in promoting compliance with early postoperative home rehabilitation.-A randomized clinical trial adequately powered is warranted to determine the clinical impact of the proposal.

The study is interesting.

I have only some minor suggestions with a pure academic spirit.

1.        The abstract must be more effective better summarizing the sections.

2.        The hypothesis must be listed before the aim

3.        Par. 2.1 “Design and ethics”. I suggest to separate it and to dedicate a paragraph to the ethics at the end of the methods

4.        Insert a flow chart in the methods to help the reader

5.        In the discussion insert a paragraph with suggestions for further work

6.        Expand a bit the conclusions

Author Response

Comments to the reviewer:

Thank you for the time spent in making a thorough review of our manuscript. We have studied your comments and suggestions. They are reasonable and we believe that they can increase the quality of the manuscript if we address them correctly. In the following lines, you may find a point-by-point response to address the issues and suggestions

Comments to the authors:

I have only some minor suggestions with a pure academic spirit.

  1. The abstract must be more effective better summarizing the sections.
    • Dear reviewer, we have worked on the abstract to provide complete summarized information, as requested.
  2. The hypothesis must be listed before the aim
    • Amended: “This research hypothesized that the use of a chatbot that communicates via an instant messaging application in a smartphone could be an effective strategy to promote adherence and clinical outcomes in patients undergoing RSA when compared to usual care. The goal was to conduct a pilot trial to assess the effects of the proposed chatbot on adherence and clinical outcomes assessed at 12 weeks after RSA
  3. 2.1 “Design and ethics”. I suggest to separate it and to dedicate a paragraph to the ethics at the end of the methods
    • Amended
  4. Insert a flow chart in the methods to help the reader
    • Included as Figure 1.
  5. In the discussion insert a paragraph with suggestions for further work.
    • We have included the following sentence, after presenting the limitations: “The results of this pilot study can guide the design of future research, and warrant clinical trials with adequate power to support the findings. These could be oriented to solve some of the mentioned limitations, with longer follow-up periods and a higher sample size.”
  6. Expand a bit the conclusions:
    • The conclusions read now: “Overall, this study suggested that a virtual assistant that interacts as a chatbot via an instant messaging application in a smartphone can be an effective strategy to promote adherence and increase compliance rates to early postoperative home rehabilitation in patients undergoing RSA. Longer follow-up periods of 6 months to 1 year and a higher sample size could help to ultimately confirm such a statement and give clinical importance to the findings. The positive results of this pilot trial can help the design of future randomized clinical trials adequately powered to determine the short- and long-term clinical impact.”

Round 2

Reviewer 1 Report

Comments and Suggestions for Authors

I have no further comments.

Comments on the Quality of English Language

There are still English quality problems and require a thorough revision of English editing.

Author Response

Dear reviewer,

The article has been edited by MPDI editing services to provide high English standards,

Thank you for your review work,

Jose